# Exploring the torque- velocity relationship in postmenopausal women: Analyzing the influence of data processing

Jessica Rial-Vázquez, Alejandra Camacho-Villa, Sonia Liliana Rivera-Mejía, Iván Nine, María Rúa-Alonso, Juan Fariñas, Borja Revuelta-Lera, Eliseo Iglesias-Soler*

Performance and Health Group, Department of Physical Education and Sport, Faculty of Sports Sciences and Physical Education, University of A Coruna, A Coruña, Spain

* eliseo.iglesias.soler@udc.es

## Abstract

### Objective

The main aims of this study were to compare the goodness of fit and derived parameters of linear and non-linear models for fitting the torque-velocity (TV) relationship in postmenopausal women, and to examine the influence of data processing on the results obtained.

### Methods

Sixteen physically active postmenopausal women completed the experiment. Knee extensor (KE) and elbow flexor (EF) muscle strength was evaluated in the dominant limb using an isokinetic dynamometer. Isometric and isokinetic tests were conducted at 30, 60, 120, 180, 240, and 300°/s. Peak torque and the corresponding joint angles were recorded for each test. TV data were fitted using linear, quadratic polynomial (PM), and Hill's (HM) regression models. TV relationships were analyzed using both actual data (i.e., the velocity achieved and its associated torque; $TV_A$) and target data (i.e., the velocity preset on the dynamometer and the torque reported; $TV_T$). TV parameters derived from each model and their goodness of fit were calculated for both $TV_A$ and $TV_T$ relationships.

### Results

The goodness of fit and the estimated TV parameters derived from the regression models differed significantly between $TV_A$ and $TV_T$ for both KE and EF ($P < 0.05$). For $TV_A$, the models with the best fit were HM for KE and PM for EF. However, HM yielded unrealistically high theoretical maximum velocity values ($6764.69 \pm 11619.09$°/s) for KE. Parameter estimates for $TV_A$ differed significantly between models ($P < 0.001$).

**Data availability statement:** All relevant data for this study are publicly available from the Zenodo repository (https://zenodo.org/records/15593957) and within the Supporting Information files.

**Funding:** This study is part of the project PID2021-124277OB-I00 funded by MCIN/AEI/ 10.13039/501100011033 and "ERDF/EU. M.R.A. and J.R.V. received financial support from the Spanish Ministry of Universities through the Grants for the Requalification of the Spanish University System under the Postdoctoral Margarita Salas Programme – Universidade da Coruña (RSUC.UDC.MS09 and RSUC.UDC.MS10, respectively). MRA also acknowledges the financial support received from the Xunta de Galicia (Consellería de Cultura, Educación, Formación Profesional y Universidades) through the Xunta de Galicia Postdoctoral Fellowships (ED481B -2024 -077). I.N. is supported by a predoctoral grant from Spanish Ministry of Science, Innovation and Universities (FPU23/03727). The funders had no role in study design, data collection and analysis, decision to publish, or preparation of the manuscript.

**Competing interests:** The authors have declared that no competing interests exist.

## Conclusion

Caution is advised when performing isokinetic assessments at high velocities in middle-aged women. The obtained data should be carefully examined, as $TV_A$ and $TV_T$ should not be used interchangeably. The choice of model can influence the estimated parameters. We recommend using quadratic polynomial models to fit TV data for both KE and EF in postmenopausal women.

## 1 Introduction

Midlife women experience a natural decline in circulating estrogen levels, which affects their health status. Hormonal changes impact body composition by increasing abdominal fat mass and decreasing both fat-free mass and bone mineral density [1]. Additionally, physical inactivity combined with aging promotes a decline in muscle mass, thereby reducing the muscles' ability to generate force [2,3]. In particular, women experience an accelerated loss of muscle mass and strength at an earlier age than men, especially with the onset of menopause [4]. In this context, evaluating strength fitness throughout life is a valuable strategy to monitor the deleterious changes associated with menopause and aging in women. The assessment of isokinetic metrics, such as torque and power, has been considered the gold-standard method for testing muscle strength due to its high levels of reliability and validity [5]. Previous studies have indicated that aging contributes to a loss of torque at high velocities (>120°/s) in knee extensor muscles, partly due to a reduction in the size and number of type II muscle fibers [6–8]. Van Roie et al. [9] established a frailty threshold at 350°/s of unloaded movement speed and 1.46 N/kg isometric force in unilateral knee extension in women over 70 years of age. Their results revealed that movement speed is a crucial factor in the onset of functional difficulties in elderly women. Therefore, evaluating torque across a wide range of angular velocities may provide a comprehensive representation of women's mechanical capabilities, summarized by the torque-angular velocity (TV) profile. Moreover, it has been suggested that not all muscle groups are equally affected by aging [10]. For example, it has been shown that knee extensors are more susceptible to age-related power loss with increasing velocities than dorsiflexor muscles [10]. Therefore, the exploration of the TV relationship in different muscle groups, including both upper and lower limbs, is recommended to better characterize the neuromuscular profile of postmenopausal women.

Several mathematical approaches can be applied to torque and velocity data to estimate the maximal capacity of active muscles to produce torque ($T_0$), and power at different angular velocities. $T_0$ is a useful parameter, commonly associated with maximum isometric torque [11,12], and serves as a reliable indicator of isometric force, offering a non-invasive alternative to direct measurement. However, this association has been scarcely explored in middle-aged women. Furthermore, the theoretical maximum angular velocity ($V_0$) is an estimated representation of the unloaded shortening velocity of the muscles involved in a specific movement. As a

theoretical construct, it may help define the range of velocities potentially achievable (i.e., velocities at which a participant can effectively exert force during the movement) within a progressive isokinetic evaluation. The classical Hill equation [13] is commonly used to model this relationship [14–16]; however, other approximation models are also currently in use. The study conducted by Magris et al. [17] observed no significant differences in the estimation of $V_0$ when using linear or linear-hyperbolic models [18] to fit the TV relationship in knee extensors. Thus, for simplicity, they opted to use a linear TV approach. Similarly, other evidence recommends that approaches to the force-velocity relationship should be based on linear models, as they provide greater reliability in parameter estimation and are simpler than quadratic polynomial, hyperbolic, or exponential models [19,20]. Given that middle-aged women experience a loss of torque at various speeds, particularly at higher velocities, it is relevant to explore which models best fit the TV relationship in this population. To the best of our knowledge, this topic has not been previously addressed.

On the other hand, accurately obtaining a TV relationship requires that data be properly recorded during the evaluation and appropriately processed afterward. In isokinetic protocols, it is recommended that torque performance be recorded during the portion of the movement in which the target velocity is achieved, excluding the acceleration and deceleration phases of the range of motion [21,22]. However, it has recently been reported that some elderly women are unable to reach target angular velocities greater than 300°/s during isokinetic knee extension exercises [22]. Therefore, the method used to process torque and velocity data, whether based on target or actual velocities, must influence the resulting TV profiles in both upper and lower limbs.

In view of the above, the main aim of this study was to compare the goodness of fit and the TV parameters obtained from linear and non-linear models using both actual ($TV_A$) and target ($TV_T$) maximum angular velocities as the independent variable in postmenopausal women, for both upper and lower limbs. Additionally, we aimed to contrast commonly derived TV parameters (i.e., $V_0$ and $T_0$). As a complementary analysis, this study also aimed to explore the association between isometric torque and $T_0$ values obtained from linear and non-linear models.

## 2 Materials and methods

Sixteen physically active postmenopausal women (age: 59.31 ± 3.28 years; body mass: 64.13 ± 10.81 kg; height: 162.81 ± 5.06 cm; BMI: 24.13 ± 3.18 kg/m²) participated in this study. The inclusion criteria were: (i) being between 55 and 65 years old, (ii) being physically active (classified as moderate or high in the Global Physical Activity Questionnaire) [23] and (iii) having experienced at least 12 consecutive months without menstruation. These women were recruited for a study designed to evaluate the acute cardiovascular responses to resistance training sessions employing different set configurations. Recruitment started on February 26, 2024, and ended on March 20, 2024. The study was registered at ClinicalTrials.gov (Clinical Trial Registration: NCT05544357).

All participants read and signed a written informed consent form prior to participating in the study. The experimental design was approved by the Territorial Research Ethics Committee of A Coruña-Ferrol (Galicia, Spain; Reference number: 2022/313) and conducted in accordance with the Declaration of Helsinki.

### 2.1 Procedures

Participants completed one familiarization session and one testing session, separated by 48 hours. The strength of the knee extensor and elbow flexor muscles was evaluated in the dominant limb using a Humac Norm isokinetic dynamometer (Humac Norm, CSMI, Boston, Massachusetts, USA). All measurements were performed by the same researcher, and the equipment was calibrated according to the manufacturer's specifications before each testing session.

### 2.2 Familiarization

Participants completed two 5-second isometric contractions and one set of two dynamic repetitions at 30°/s and 300°/s for both knee extension and elbow flexion exercises, with 1 minute of rest between sets and 2 minutes between conditions

and exercises. Additionally, the chair position, dynamometer placement, and the length of the attachment arm were individually adjusted to ensure proper alignment of the anatomical axes of the knee and elbow joints with the rotational axis of the lever arm.

## 2.3 Testing sessions

The session began with a standardized warm-up on a stationary bicycle (Monark 828E; Monark Exercise AB, Vansbro, Sweden) for five minutes at approximately 70 rpm. Strength assessments were then conducted using the dynamometer, starting with the knee extensor exercise, followed by the elbow flexor exercise. The protocol for each exercise is detailed in the following sections.

### 2.3.1 Unilateral knee extensor strength assessment.
Participants were seated with hips flexed at 100° and knees at 90°. To minimize positional shifts, stabilization straps were placed around the chest, pelvis, and thigh of the tested leg. The shank was secured to the dynamometer approximately 3 cm above the medial malleolus, and the mechanical axis of the lever arm was aligned with the lateral epicondyle of the knee. The range of motion (ROM) for all isokinetic contractions was set from 80° (knee flexed) to 0° (full extension). The maximal isometric contraction was performed at 60° of knee flexion, and gravity correction was applied with the knee stabilized in full extension (0°). Illustrative details of the setup are provided in Fig 1.

As part of the warm-up, participants performed three submaximal 5-second contractions, with 60 seconds of rest between each. Subsequently, three maximal isometric contractions were executed under the same conditions.

After a two-minute rest, isokinetic testing was performed at six angular velocities: 30°, 60°, 120°, 180°, 240°, and 300°/s. Prior to each maximal effort, participants completed three incremental submaximal repetitions. After a 60-second rest, three maximal isokinetic contractions were performed at each velocity, with 30 seconds of rest between repetitions and two minutes between velocity conditions. Participants were instructed to move as quickly and forcefully as possible throughout the full ROM.

### 2.3.2 Unilateral elbow flexor strength assessment.
Participants were positioned supine, with hips flexed at 90° and knees at 120°. Stabilization straps were applied horizontally over the shoulders, chest, pelvis, and arm of the tested limb. The elbow joint was aligned with the axis of the dynamometer, and the forearm was maintained in a neutral position. The ROM was set from 20° (near full extension) to 120° (elbow flexed). The maximal isometric contraction was performed at 90° of elbow flexion (with 0° corresponding to full extension). A visual representation of the setup is provided in Fig 1.

As in the lower-limb protocol, participants first completed three submaximal 5-second contractions, with a 60-second rest between each. This was followed by three maximal isometric contractions, with standardized verbal encouragement to maximize effort.

After a two-minute rest, isokinetic testing was performed at six angular velocities: 30°, 60°, 120°, 180°, 240°, and 300°/s. Prior to each velocity condition, participants performed three incremental submaximal repetitions without rest. After a 60-second rest, three maximal isokinetic contractions were carried out at each velocity, with 30 seconds of rest between repetitions and two minutes between conditions. Participants were instructed to move as fast and forcefully as possible throughout the full ROM.

## 2.4 Data analysis

Torque during each maximal isometric contraction was quantified as the peak value attained over a 5-second interval. The best score from the three attempts was retained for further analysis. For isokinetic evaluation, peak torque values were obtained from the best trial at each angular velocity (30°/s, 60°/s, 120°/s, 180°/s, 240°/s, and 300°/s). The TV relationship was derived for each participant based on the range of angular velocities tested and the corresponding peak torque values.

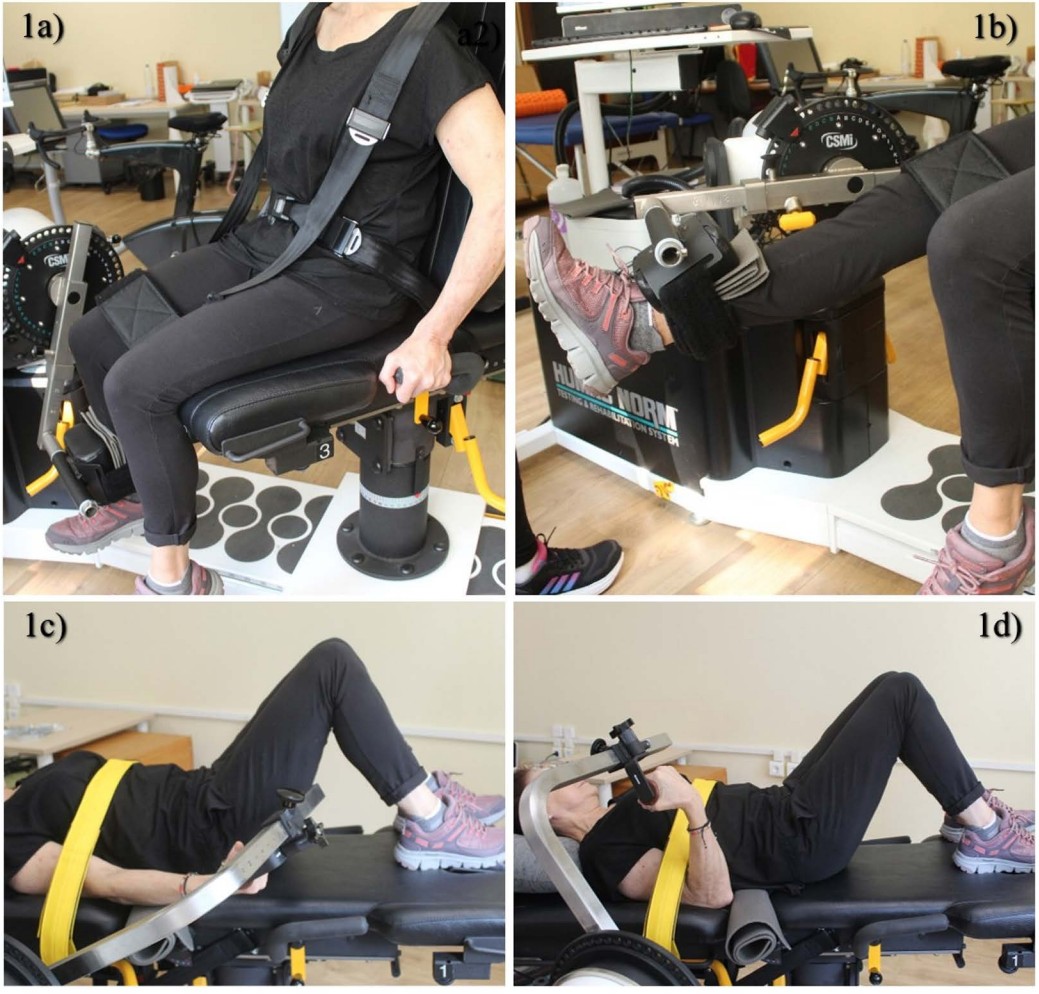

**Fig 1. Isokinetic experimental setup.** 1a: Initial position (80°) during the execution of unilateral knee extension exercise; 1b: final position (0°) during the execution of unilateral knee extension exercise; 1c: Starting position (20°) for performing unilateral elbow flexion; 1d: final position (120°) for unilateral elbow flexion exercise.

Since it was observed that the dynamometer provided peak torque values even when the target angular velocity had not been reached, the TV relationship was computed using both actual (i.e., the velocity actually achieved and its associated torque; $TV_A$) and target data (i.e., the preset velocity on the dynamometer and the reported torque; $TV_T$). To obtain the $TV_A$ relationship for both lower and upper limbs, peak torque was identified from the portion of the contraction in which >90% of the maximum angular velocity achieved was reached during each condition. The angular velocity at which this peak torque occurred was used for analysis. If peak torque was reached at a lower velocity than in the previous condition, the data point was excluded from analysis. Additionally, the joint angle corresponding to each peak torque was recorded for all velocity conditions. For the $TV_T$ relationship, the angular velocity and peak torque values were directly extracted from the dynamometer report.

When fitting the TV relationship using the linear model (LM), we estimated the theoretical maximum torque at null angular velocity ($T_0$), the theoretical maximum angular velocity ($V_0$), and the slope of the relationship ($S_{TV}$: $-V_0/T_0$). In the case of the second order polynomial model (PM), the TV relationship was expressed as:

$$T = a \cdot V^2 + b \cdot V + c$$

where $T$ and $V$ represent torque and angular velocity, respectively. In this model, the intercept $c$ corresponds to $T_0$, and $V_0$ is estimated as the velocity at which $T$ equals zero. Since a second-order polynomial has two roots (i.e., two $V$ values at which $T=0$), $T_0$ was derived from the root where the function is decreasing (i.e., where the first derivative is negative). The coefficient $a$ corresponds to the second derivative of the function and reflects the concavity of the curve. For the Hill's model (HM) [13], the TV relationship was expressed as:

$$(T + a)(V + b) = (T_0 + a)\,b.$$

where $T$ and $V$ are torque and velocity, respectively; $T_0$ represents isometric torque; $a$ is a constant reflecting the shortening heat per unit of shortening (with the dimensions of torque); and $b$ is a constant that reflects the increase in energy rate per unit of decrease in torque (with the dimensions of velocity).

## 2.5  Statistical analysis

Descriptive data are presented as means±standard deviations, except for the coefficient of determination ($R^2$), which is reported as mean and range. The normality of the data was assessed using the Shapiro–Wilk test. When the assumption of normality was violated, non-parametric tests were applied.

The coefficient of determination ($R^2$), mean square error (MSE) — as indicators of goodness of fit — and all TV parameters estimated from each model were compared using both $TV_A$ and $TV_T$ data. When data were normally distributed, paired t-tests were used to compare both approaches. Hedges' g and the corresponding 95% confidence intervals (CI) were calculated to assess effect sizes. Thresholds for interpreting effect sizes were set at 0.2 (small), 0.5 (medium), and 0.8 (large). For non-parametric comparisons, the Wilcoxon signed-rank test was employed, and the effect size was calculated using rank biserial correlation. This coefficient ranges from −1–1, where −1 indicates that all observations in the second condition are greater than those in the first, 1 indicates the opposite, and 0 reflects no systematic difference between conditions.

In seven cases, $V_0$ could not be estimated using the Hill model: two in $TV_A$, two in $TV_T$ for knee extensors (KE), and three in $TV_T$ for elbow flexors (EF).

Subsequent analyses were conducted using $TV_a$ data only. Specifically, $R^2$ and MSE from individual $TV_A$ regressions were compared across the linear (LM), polynomial (PM), and Hill (HM) models using Friedman's test, with effect size expressed as Kendall's W. This coefficient indicates the degree of agreement in rankings across subjects, where a value of 1 denotes perfect agreement and 0 represents random ranking. When Friedman's test was significant, pairwise Wilcoxon signed-rank tests with Bonferroni correction were performed as post hoc analysis.

To compare the derived TV parameters ($V_0$ and $T_0$) between models (LM, PM, HM), a one-way ANOVA was conducted with "model" as a fixed factor.

To evaluate concordance between the maximal isometric torque and the $T_0$ derived from the LM, PM, and HM, a one-way ANOVA was used to assess systematic bias. Additionally, Pearson's correlation coefficients (r) were calculated to assess associations between measured and model-derived values. Lin's concordance correlation coefficient was also computed by multiplying Pearson's r by a coefficient of bias (c.b), which reflects the deviation of the regression line from the 45° identity line [24]. The coefficient of bias ranges from 0 to 1, with values near 1 indicating high agreement. Thus, Lin's coefficient captures both precision and accuracy.

Finally, a one-way ANOVA with "velocity" as the factor was used to compare the joint angles associated with peak torque across all velocity conditions (30, 60, 120, 180, 240, and 300°/s). Effect sizes for all ANOVAs were reported as partial eta squared ($\eta^2$). Post hoc pairwise comparisons were conducted with Bonferroni adjustments, when appropriate.

All statistical analyses were performed using IBM SPSS Statistics v27.0 (IBM Corp., Armonk, NY, USA), GraphPad Prism v5.01 (GraphPad Software, San Diego, CA, USA), and the esc and effect size packages in RStudio (version 2024.12.1).

## 3 Results

The goodness of fit indicators ($R^2$ and MSE) and the estimated torque–velocity (TV) parameters from each model were compared between $TV_A$ and $TV_T$ approaches (Table 1).

For knee extension (KE), both $R^2$ and $T_0$ derived from the linear model (LM) and the polynomial model (PM) were lower when using $TV_T$ compared to $TV_A$. Conversely, $V_0$ values from the LM were higher when calculated with $TV_T$ data.

In the case of elbow flexion (EF), $R^2$ and $T_0$ values obtained from the LM were also lower under the $TV_t$ condition, while $V_0$ was higher compared to the $TV_A$ condition. The goodness of fit for PM and Hill's model (HM) showed minimal differences when comparing $TV_T$ and $TV_A$. However, PM resulted in higher $T_0$ values when $TV_T$ data were used. Regarding HM, lower $T_0$ and higher $V_0$ values were observed under $TV_T$ compared to $TV_A$.

Since significant differences were observed between $TV_T$ and $TV_A$ regardless of the regression model applied, all subsequent analyses were conducted using $TV_A$ data. Table 2 summarizes the goodness of fit and the commonly derived TV parameters across models.

Comparisons between the maximum isometric torque and the estimated $T_0$ values are illustrated in Fig 2. Pearson correlation coefficients and Lin's concordance correlation coefficients for these variables are presented in Table 3.

Additionally, Fig 3 displays the joint angles at which peak torque occurred for each velocity condition during the isokinetic test.

## 4 Discussion

The main findings of this study were: (a) overall, the goodness of fit and the estimated torque-velocity parameters derived from regression models differed when comparing $TV_A$ and $TV_T$ for both KE and EF; (b) regarding $TV_A$, the models with the best goodness of fit were HM for KE and PM for EF; (c) the estimation of $TV_A$ parameters for KE and EF varied depending on the model used; (d) for KE, isometric peak torque and $T_0$ were similar when $T_0$ was obtained from PM or HM, but differed when using LM. In contrast, for EF, isometric peak torque and $T_0$ varied across all models.

After identifying issues and inconsistencies in the obtained data, a thorough review of the data processing was carried out. The isokinetic evaluation included errors in assigning peak torque values to incorrect velocities, particularly from the 180°/s condition, and notably in the elbow flexion exercise. A basic requirement during isokinetic investigations is that the preset velocity should be maintained over most of the considered ROM. Handel et al. [24] reported that an isokinetic dynamometer must first be accelerated to the preset velocity by the subject's muscle force. If the initial applied force is insufficient, the acceleration phase may be prolonged, delaying the achievement of target angular velocities. This reduces the range of angles considered, since peak torque should be obtained from the portion of movement where target velocity is maintained. For velocities greater than 180°/s, the isokinetic evaluation report (i.e., exported directly from Humac software v.9.7.1) did not always exclude the acceleration or deceleration phases, providing peak torque values at angular velocities much lower than those planned.

Specifically, 2.1% of KE and 20.8% of EF peak torques were incorrectly identified by the Humac software. In the records where peak torque was correctly identified, the associated velocity consistently exceeded 90% of the target velocity but did not always match it precisely. For KE, this deviation reached up to 6% at 240°/s and 8% at 300°/s. In EF, five participants (31% of the sample) failed to achieve peak torque at a velocity reaching at least 90% of the target condition at 240°/s and 300°/s. Consequently, peak torque was determined from the portion where velocities exceeded 90% of the maximum achieved velocity for those individuals. For instance, in the 300°/s condition, peak torque for two participants was recorded at 225°/s and 211°/s, corresponding to maximum achieved velocities of 250°/s and 240°/s, respectively.

**Table 1. Goodness of fit of each regression model of the torque velocity relationship considering both actual and target angular velocity as independent variable. Coefficients of determination are presented as means and ranges while mean square errors and torque velocity parameters are represented as means± standard deviations.**

| Exercise | Model | Parameter | $TV_A$ | $TV_T$ | Wilcoxon or Paired T-test* | Rank biserial correlation or Hedge's G* (95% CI) |
|---|---|---|---|---|---|---|
| **KE** | LM | $R^2$ | 0.929 (0.843-0.975) | 0.916 (0.822-970) | P=0.001 | 0.93 (0.79, 0.98) |
| | | MSE ($N^2 \cdot m^2$) | 10.53±4.11 | 11.43±4.55 | P=0.003 | -0.85 (-0.95, -0.61) |
| | | $T_0$ (N·m) | 140.09±28.17 | 138.43±27.65 | P<0.001* | 1.20* (0.47, 1.93) |
| | | $V_0$ (°/s) | 411.82±51.69 | 437.81±55.95 | P<0.001 | -1.00 (1.00, 1.00) |
| | | $S_{TV}$ (N·m·s/°) | -0.35±0.10 | -0.32±0.09 | P<0.001* | -2.70* (-3.64, -1.76) |
| | PM | $R^2$ | 0.989 (0.953-0.999) | 0.987 (0.948-0.998) | P=0.017 | 0.68 (0.26, 0.88) |
| | | MSE ($N^2 \cdot m^2$) | 4.02±2.15 | 4.36±2.39 | P=0.023 | -0.65 (-0.87, -0.21) |
| | | $T_0$ (N·m) | 159.90±34.78 | 159.01±34.65 | P=0.008* | 0.56* (-0.13, 1.24) |
| | | $V_0$ (°/s) | 311.04±61.96 | 323.45±74.55 | P=0.215 | -0.35 (-0.73, 0.19) |
| | | $X^2$ | $1.27 \times 10^{-3} \pm 0.55 \times 10^{-3}$ | $1.22 \times 10^{-3} \pm 0.51 \times 10^{-3}$ | P=0.027* | 0.45* (-0.23, 1.13) |
| | | x | -0.74±0.26 | -0.71±0.26 | P=0.018* | -0.80* (-1.50, 0.10) |
| | HM | $R^2$ | 0.988 (0.970-0.999) | 0.989 (0.963-0.998) | P=0.179 | 0.38 (-0.15, 0.74) |
| | | MSE ($N^2 \cdot m^2$) | 4.07±2.31 | 4.22±1.84 | P=0.070 | -0.51 (-0.81, -0.01) |
| | | $T_0$ (N·m) | 171.79±42.34 | 171.89±42.14 | P=0.940* | -0.01* (-0.68, 0.67) |
| | | $V_0$ (°/s) | 6764.69±11619.09 | 16732.40±40351.96 | P=0.308 | -0.33 (-0.76, 0.29) |
| | | a | 19.31±23.51 | 23.51±35.07 | P=0.654* | -0.11* (-0.78, 0.57) |
| | | b | 182.33±98.50 | 180.27±101.76 | P=0.831* | 0.03* (-0.65, 0.70) |

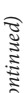

*(Continued)*

**Table 1.** (Continued)

| Exercise | Model | Parameter | $TV_A$ | $TV_T$ | Wilcoxon or Paired T-test* | Rank biserial correlation or Hedge´s G* (95% CI) |
|---|---|---|---|---|---|---|
| EF | | $R^2$ | 0.957 (0.823-0.995) | 0.852 (0.343-0.996) | P=0.023 | 0.65 (0.21, 0.87) |
| | | MSE ($N^2 \cdot m^2$) | 1.38±0.74 | 1.88±1.22 | P=0.098 | −0.47 (−0.79, 0.05) |
| | LM | $T_0$ (N·m) | 30.22±5.17 | 28.42±4.60 | P<0.001* | 1.22* (0.46, 1.97) |
| | | $V_0$ (°/s) | 481.93±163.59 | 649.59±175.77 | P<0.001 | 1.00 (1.00, 1.00) |
| | | $S_{TV}$ (N·m·s/°) | −0.07±0.02 | −0.05±0.01 | P<0.001* | 1.27* (0.50, 2.03) |
| | PM | $R^2$ | 0.975 (0.912-0.999) | 0.973 (0.924-0.997) | P=0.717 | 0.10 (−0.43, 0.56) |
| | | MSE ($N^2 \cdot m^2$) | 1.31±0.92 | 1.03±0.64 | P=0.070 | 0.51 (0.01, 0.81) |
| | | $T_0$ (N·m) | 29.32±5.58 | 30.82±6.34 | P<0.001* | −1.19* (−1.95, −0.44) |
| | | $V_0$ (°/s) | 421.19±85.95 | 376.37±138.14 | P=0.352 | 0.26 (−0.28, 0.68) |
| | | $X^2$ | $-0.56 \times 10^{-4} \pm 0.90 \times 10^{-4}$ | $1.39 \times 10^{-4} \pm 1.79 \times 10^{-4}$ | P<0.001* | −1.37* (−2.15, −0.60) |
| | | X | −0.05±0.04 | −0.09±0.06 | P<0.001* | 1.26* (−2.02, −0.50) |
| | HM | $R^2$ | 0.940 (0.793-0.991) | 0.889 (0.437-0.995) | P=0.836 | 0.06 (−0.46, 0.55) |
| | | MSE ($N^2 \cdot m^2$) | 1.84±0.68 | 1.76±1.07 | P=0.196 | 0.37 (−0.17, 0.74) |
| | | $T_0$ (N·m) | 31.59±5.84 | 29.67±4.75 | P<0.001* | 1.27* (0.50, 2.03) |
| | | $V_0$ (°/s) | 713.75±324.89 | 902.90±283.881 | P=0.002 | −0.96 (−0.99, −0.86) |
| | | a | 30.89±6.70 | 23.95±12.79 | P=0.040* | 0.57* (−0.14, 1.28) |
| | | b | 693.64±326.49 | 799.73±320.28 | P=0.012* | −0.76* (−1.48, −0.04) |

KE: knee extension; EF: elbow flexion; LM: linear model; PM: polynomial model; HM: Hill model; TV: Torque-velocity relationship; $TV_A$: Torque-velocity relationship considering actual angular velocity as independent variable; $TV_T$: Torque-velocity relationship considering target angular velocity as independent variable; $T_0$: theoretical maximum torque; $V_0$: theoretical maximum velocity; $S_{TV}$: slope of the linear TV relationship; $R^2$: coefficient of determination; MSE: mean square errors; *: the analysis was performed with paired t-test and effect sizes are reported with Hedges'G.

**Table 2. Comparison of the goodness of fit and shared torque–velocity parameters derived from each regression model. Coefficients of determination ($R^2$) are reported as means and ranges, while mean square errors (MSE) are presented as means±standard deviations.**

| Exercise | Parameter | LM | PM | HM | One-way ANOVA or Friedman analysis | Kendall´s W or η² |
|---|---|---|---|---|---|---|
| **KE** | $R^2$ | 0.929 #* (0.843-0.975) | 0.989 (0.953-0.999) | 0.991 (0.969-0.999) | p<0.001 | 0.71 |
| | MSE (N²·m²) | 10.53±4.11#* | 4.02±2.15# | 3.72±1.63 | p<0.001 | 0.66 |
| | $T_0$ (N·m) | 140.09±28.17#* | 159.90±34.78# | 171.79±42.34 | p<0.001 | 0.66 |
| | $V_0$ (°/s) | 411.82±51.69#* | 311.04±61.96# | 6764.69±11619.09 | p<0.001 | 1.00 |
| **EF** | $R^2$ | 0.957#* (0.823-0.995) | 0.975# (0.912-0.999) | 0.940 (0.793-0.991) | p<0.001 | 0.78 |
| | MSE (N²·m²) | 1.38±0.74#* | 1.31±0.93 | 1.93±0.87 | p<0.001 | 0.59 |
| | $T_0$ (N·m) | 30.22±5.17#* | 29.32±5.58# | 31.59±5.84 | p<0.001 | 0.71 |
| | $V_0$ (°/s) | 481.93±163.59#* | 421.00±85.95# | 713.75±324.89 | p<0.001 | 0.81 |

KE: knee extension exercise; EF: elbow flexion exercise; LM: linear model; PM: polynomial model; HM: Hill's model; $T_0$: theoretical maximum torque; $V_0$: theoretical maximum velocity; $R^2$: coefficient of determination; MSE: mean errors; #: significant differences in comparison with HM (p<0.05); *: significant differences in comparison with PM (p=≤0.001). Kendall's W: Kendall's coefficient of concordance; η²: effect size by eta square.

One-way ANOVA was used for $T_0$ in both exercise and Friedman's for the the rest of the parameters..

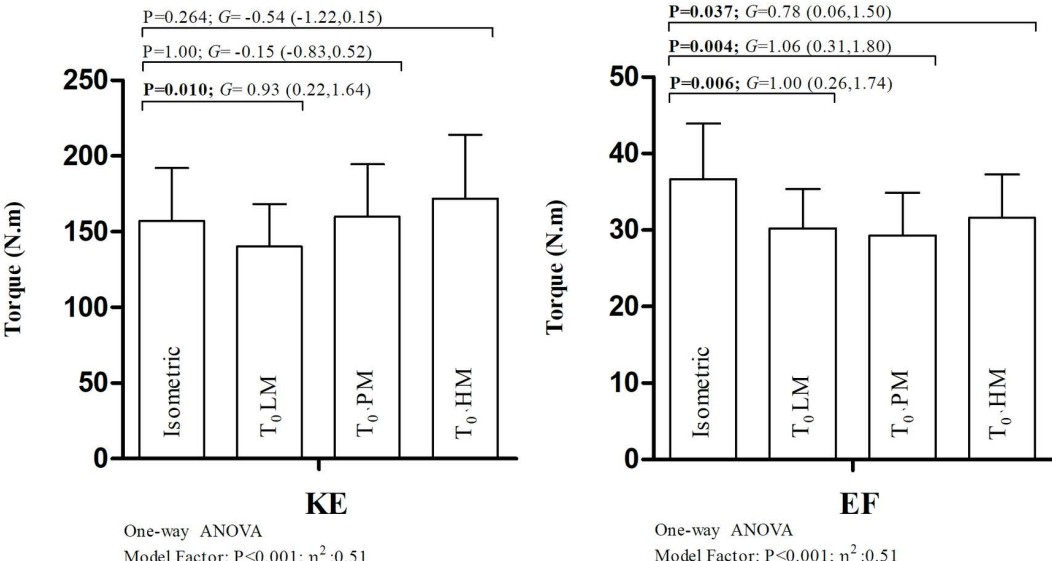

**Fig 2. One-way ANOVA results comparing peak isometric torque and the theoretical peak torque values estimated by the linear, quadratic polynomial, and Hill's models.** KE: knee extension exercise; EF: elbow flexion exercise; Isometric: maximum measured peak torque; $T_0$: theoretical maximum torque; LM: linear model; PM: polynomial model; HM: Hill's model; η²: partial eta squared; G: Hedges' g and the corresponding 95% confidence intervals (CI).

Therefore, we ultimately chose to analyze the raw data to correctly assign peak torques to an actual velocity value that at least met the requirement of being >90% of the achieved velocity. The observed inconsistencies may be influenced by the population profile. Prior literature indicates that aging contributes to a loss of torque at high velocities, particularly at 120°/s and above [8]. Additionally, a previous study [22] confirmed that several elderly women (>61 years) were unable to achieve contraction velocities >300°/s in KE. Compared with younger individuals, older adults demonstrate a slower rate of twitch torque development, longer relaxation times, and greater impairments in torque- and power-generating capacity

**Table 3. Pearson correlation coefficients and Lin's concordance correlation coefficients between the measured isometric peak torque and the theoretical maximum torque ($T_0$) estimated using linear, polynomial, and Hill models.**

| | | r | Rho | S.Shift | L.Shift | c.b |
|---|---|---|---|---|---|---|
| KE | LM | 0.879 P<0.001 | 0.749 (0.480-0.890) | 0.803 | −0.517 | 0.864 |
| | PM | 0.843 P<0.001 | 0.844 (0.600-0.944) | 0.984 | 0.086 | 0.996 |
| | HM | 0.704 P=0.003 | 0.716 (0.376-0.886) | 1.159 | 0.350 | 0.933 |
| EF | LM | 0.532 P=0.034 | 0.322 (0.009-0.579) | 0.711 | −1.087 | 0.606 |
| | PM | 0.468 P=0.068 | 0.268 (−0.034-0.526) | 0.768 | −1.191 | 0.574 |
| | HM | 0.526 P=0.036 | 0.386 (0.017-0.663) | 0.782 | −0.814 | 0.734 |

KE: knee extension exercise; EF: Elbow flexion exercise; Isometric: maximum peak torque; $T_0$: theoretical maximum torque; LM: linear model; PM: polynomial model; HM: Hill's model; r: Pearson correlation coefficient; Rho: Rho correlation coefficient; S.Shift: Systematic Shift; L.Shift: Location Shift; c.b: concordance bias.

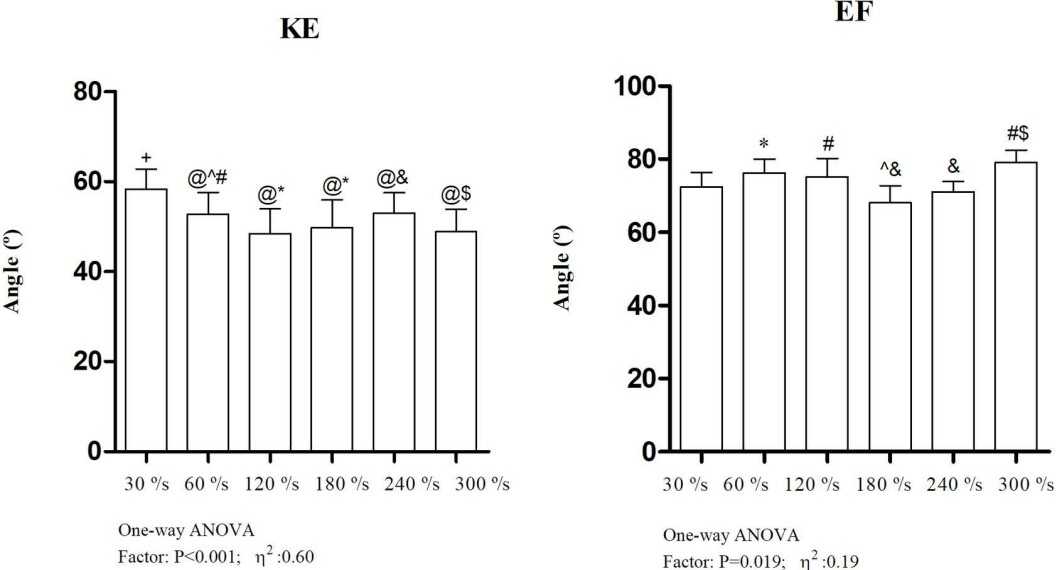

Fig 3. One-way ANOVA results for the joint angles at which peak torque was achieved at each isokinetic velocity. KE: knee extension; EF: elbow flexion; +: different from all the velocity conditions P<0.05; @: different from 30°/s P<0.05; *: different from 60°/s P<0.05; ^: different from 120°/s p<0.05; #: different from 180°/s p<0.05; $: different from 240°/s P<0.05; &: different from 300°/s P<0.05. $\eta^2$: partial eta square.

at faster velocities—likely due to a smaller proportional area of fast-twitch fibers. Moreover, the onset of menopause is associated with accelerated loss of muscle mass and strength in women [4]. Although our participants were physically active, their force levels were low, particularly in the upper limbs. These factors, along with the lack of familiarization with the testing protocol, may explain some of the recording and processing issues. Thus, caution is warranted when performing isokinetic assessments at high velocities in populations with low strength levels, as significant errors may arise if data is not properly processed prior to analysis. Major mistakes can be made if data is not properly process before analysis. In

this regard, in the next paragraph, we discuss the differences between analyzing the TV relationship using raw data (i.e., $TV_A$) versus using values directly provided in the report (i.e., $TV_T$).

For KE, the mean goodness of fit for linear, polynomial, and Hill's models when fitting $TV_A$ and $TV_T$ data was excellent ($R^2 \geq 0.915$). Nonetheless, significant differences were observed in both goodness of fit values and parameter estimates from the linear and polynomial models, indicating that $TV_A$ and TVT are not equivalent. Conversely, Hill's model showed consistent goodness of fit and derived parameters across both $TV_A$ and $TV_T$.

For EF, the mean goodness of fit for all models (linear and non-linear) was also excellent ($R^2 \geq 0.940$). However, differences were detected in the fit and parameter estimates derived from linear and polynomial models, again suggesting that $TV_A$ and $TV_T$ cannot be considered equivalent. While the goodness of fit of Hill's model remained similar between $TV_A$ and $TV_T$ data, all estimated parameters differed. Thus, $TV_A$ and $TV_T$ differ independently of the model used and should not be used interchangeably. We therefore recommend a meticulous review of isokinetic data to obtain an accurate TV relationship. Substantial errors can occur if peak torque is not identified correctly and near the target velocity, especially when fitting linear models. As shown in EF, the goodness of fit of linear models using $TV_T$ data can fall to poor levels ($R^2 < 0.400$), limiting the interpretability of intervention studies focused on adaptations.

In KE, the best fit was obtained using Hill's model ($R^2 = 0.969$–$0.999$), and the worst with LM ($R^2 = 0.843$–$0.975$). However, Hill's model produced an implausible $V_0$ value (6764.69°/s), and in two cases was unable to provide $V_0$ at all. Due to these findings, we do not recommend using Hill's model to fit TV data in postmenopausal women. Herskind et al. [15] also reported unrealistic $V_0$ values (>2000°/s) for KE in young, healthy, physically active individuals using Hill's model. Despite the population's characteristics, five out of 64 curves in their study failed to provide $V_0$ values. They noted difficulties assessing the TV relationship during fast contractions due to the limited time window for achieving stable torque and velocity, leading to measurement artifacts and contributing to $V_0$ variability. Although Hill's model is frequently used for KE [14–16], other studies have employed polynomial [12] or linear models [12,17,25–27]. For instance, Bozic et al. [27] used linear models and obtained good fit ($R^2 = 0.74$–$0.97$), though their velocity range was narrower (30°/s to 180°/s). Based on previous works conducted by Grbic et al. [11], Sašek et al. [12] assessed 9 data points from 30°/s to 300°/s and found that the polynomial model had significantly better fit than linear models, despite both achieving high $R^2$. They also validated a two-point method (e.g., 30°/s with 210°/s or 240°/s or 300°/s) for rapid TV analysis. Based on our data and the literature, we recommend using the polynomial model due to its strong fit ($R^2 = 0.953$–$0.999$; MSE = $4.02 \pm 2.15$ N²·m²) and physiologically coherent parameters. Furthermore, the PM is considered a simple regression model that enables efficient parameter extrapolation. Importantly, it allows researchers to analyze the concavity of the torque–velocity curve, which has been proposed as a marker of muscle fiber composition and acute fatigue [28,29].

To our knowledge, the TV relationship in upper limbs is rarely studied [30,31], and even less so in middle-aged women [32,33]. In our study, the best fit for EF was obtained using the polynomial model ($R^2 = 0.912$–$0.999$; MSE = $1.31 \pm 0.93$ N²·m²), and the worst with Hill's model ($R^2 = 0.793$–$0.991$). Similar to KE, Hill's model could not estimate $V_0$ in three cases. Janicijevic et al. [31] analyzed EF force-velocity relationship using 8 data points (30–240°/s) and validated a two-point method (60–180°/s), reporting better linear fit ($R^2 > 0.969$) than ours ($R^2 = 0.823$–$0.995$). Pousson et al. [32] compared EF peak torques in young vs. old women at four velocities (60–240°/s) but did not estimate TV parameters, preventing comparison with our results. Given that model selection significantly impacts parameter estimates, it is a critical methodological choice. Considering the PM's significantly superior fit ($R^2 = 0.912$–$0.999$; MSE = $1.31 \pm 0.93$ N²·m²) relative to other models, along with its ability to yield meaningful insights into the curvature of the torque–velocity relationship, we recommend its application in postmenopausal women.

For KE, isometric peak torque and $T_0$ values were comparable when estimated from PM and HM but differed when derived from LM. Although LM yielded a strong Pearson correlation between $T_0$ and isometric torque ($r = 0.879$), a t-test revealed a systematic bias ($p = 0.004$), with LM underestimating $T_0$. Previous studies have reported strong associations between isometric torque and $T_0$ using LM, but typically relied solely on Pearson correlations. Grbic et al. [11] found strong

associations ($r = 0.80$–$0.84$) between KE isometric torque at 60° and $T_0$ estimated from linear models using five or two points. Sašek et al. [12], however, reported moderate associations and consistent underestimation of $T_0$ versus isometric torque, with better accuracy when using PM than LM. Our results also support this: *c.b* was closer to 1 for PM (0.996) and HM (0.933) than for LM (0.864), suggesting less deviation from the identity line between $T_0$ and isometric torque. Hence, simple correlations are insufficient; concordance should also be considered. Our findings provide additional reasons not to use LM to model the TV relationship in KE among postmenopausal women.

For EF, isometric torque and $T_0$ values varied across all models. Pearson coefficients were <0.526 for LM, PM, and HM, and t-tests confirmed significant differences ($p < 0.007$). Lin's concordance coefficients were low (<0.386), indicating poor agreement. Additionally, *c.b* values deviated notably from the identity line (range: 0.574–0.734). These discrepancies led us to reconsider the angle used for isometric testing. Analysis of peak torque angles during isokinetic EF testing revealed consistent deviation from the 90° angle used during isometric testing (see Fig 3). Another study reported higher EF peak torque at 65° compared to 90° during various isokinetic velocities (60–300°/s) [34]. Thus, future studies involving middle-aged women should consider evaluating isometric EF around 70° to better capture the angle of maximal force.

This study is not without limitations. The small sample size is a constraint, though it provides valuable insights into an underrepresented population. Additionally, measuring EF isometric strength at a single angle (90°) may have underestimated true maximum torque. Finally, the population profile is both a strength and limitation: although it provides key information for test selection, it restricts the use of gold-standard tests like isokinetic evaluations due to low strength levels [35].

In conclusion, caution should be exercised when performing isokinetic assessments at high velocities in middle-aged women. Raw data should be carefully reviewed, as $TV_A$ and $TV_T$ cannot be used interchangeably. Furthermore, the model used to fit the data significantly influences the parameter estimates. We recommend using polynomial models to fit the TV data for KE and EF in postmenopausal women.

## Supporting information

**S1 File. Data base elbow flexion supplementary information.**
(XLSX)

**S2 File. Data base knee extension supplementary information.**
(XLSX)

## Acknowledgments

We sincerely thank all the participants in this study for their effort and commitment to the study. We truly appreciate the time, patience, and enthusiasm they have shown in being part of the CARE project.

## Author contributions

**Data curation:** Jessica Rial-Vázquez, Iván Nine.

**Formal analysis:** Jessica Rial-Vázquez, Eliseo Iglesias-Soler.

**Investigation:** Alejandra Camacho-Villa, Sonia Liliana Rivera-Mejía.

**Supervision:** María Rúa-Alonso, Eliseo Iglesias-Soler.

**Visualization:** Jessica Rial-Vázquez, Iván Nine, María Rúa-Alonso, Juan Fariñas, Eliseo Iglesias-Soler.

**Writing – original draft:** Jessica Rial-Vázquez.

**Writing – review & editing:** Jessica Rial-Vázquez, Juan Fariñas, Borja Revuelta-Lera, Eliseo Iglesias-Soler.

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
