## [Decision Letter · Decision Letter 0]

PONE-D-25-21537Exploring the Torque- Velocity Relationship in Postmenopausal Women: Analyzing the Influence of Data ProcessingPLOS ONE

Dear Dr. Iglesias-Soler,

Thank you for submitting your manuscript to PLOS ONE. After careful consideration, we feel that it has merit but does not fully meet PLOS ONE’s publication criteria as it currently stands. Therefore, we invite you to submit a revised version of the manuscript that addresses the points raised during the review process.

We look forward to receiving your revised manuscript.

Kind regards,

Hasan Sozen

Academic Editor

PLOS ONE

Journal Requirements:

“This study is part of the project PID2021-124277OB-I00 funded by MCIN/AEI/ 10.13039/501100011033 and “ERDF/EU. M.R.A. and J.R.V. received financial support from the Spanish Ministry of Universities through the Grants for the Requalification of the Spanish University System under the Postdoctoral Margarita Salas Programme – Universidade da Coruña (RSUC.UDC.MS09 and RSUC.UDC.MS10, respectively). MRA also acknowledges the financial support received from the Xunta de Galicia (Consellería de Cultura, Educación, Formación Profesional y Universidades) through the Xunta de Galicia Postdoctoral Fellowships (ED481B ‐2024 ‐077). I.N. is supported by a predoctoral grant from Spanish Ministry of Science, Innovation and Universities (FPU23/03727). “

“This study is part of the project PID2021-124277OB-I00 funded by MCIN/AEI/ 10.13039/501100011033 and “ERDF/EU. M.R.A. and J.R.V. received financial support from the Spanish Ministry of Universities through the Grants for the Requalification of the Spanish University System under the Postdoctoral Margarita Salas Programme – Universidade da Coruña (RSUC.UDC.MS09 and RSUC.UDC.MS10, respectively). MRA also acknowledges the financial support received from the Xunta de Galicia (Consellería de Cultura, Educación, Formación Profesional y Universidades) through the Xunta de Galicia Postdoctoral Fellowships (ED481B ‐2024 ‐077). I.N. is supported by a predoctoral grant from Spanish Ministry of Science, Innovation and Universities (FPU23/03727). Finally, we sincerely thank all the participants in this study for their effort and commitment to the study. We truly appreciate the time, patience, and enthusiasm they have shown in being part of the CARE project.”

“This study is part of the project PID2021-124277OB-I00 funded by MCIN/AEI/ 10.13039/501100011033 and “ERDF/EU. M.R.A. and J.R.V. received financial support from the Spanish Ministry of Universities through the Grants for the Requalification of the Spanish University System under the Postdoctoral Margarita Salas Programme – Universidade da Coruña (RSUC.UDC.MS09 and RSUC.UDC.MS10, respectively). MRA also acknowledges the financial support received from the Xunta de Galicia (Consellería de Cultura, Educación, Formación Profesional y Universidades) through the Xunta de Galicia Postdoctoral Fellowships (ED481B ‐2024 ‐077). I.N. is supported by a predoctoral grant from Spanish Ministry of Science, Innovation and Universities (FPU23/03727). “

4. In the online submission form, you indicated that “The data are available on request from the author”

Reviewers' comments:

Reviewer's Responses to Questions

**Comments to the Author**

1. Is the manuscript technically sound, and do the data support the conclusions?

Reviewer #1: Yes

Reviewer #2: Yes

Reviewer #3: Yes

2. Has the statistical analysis been performed appropriately and rigorously? 

Reviewer #1: Yes

Reviewer #2: Yes

Reviewer #3: Yes

3. Have the authors made all data underlying the findings in their manuscript fully available?

Reviewer #1: No

Reviewer #2: No

Reviewer #3: No

4. Is the manuscript presented in an intelligible fashion and written in standard English?

Reviewer #1: Yes

Reviewer #2: Yes

Reviewer #3: Yes

5. Review Comments to the Author

Reviewer #1: In the article "Exploring the Torque- Velocity Relationship in Postmenopausal Women: Analyzing the Influence of Data Processing" authors discussed the linear and non-linear models for fitting the torque-velocity relationship in postmenopausal women. The article is well supported with faithful and adequate results. Hence, recommended for acceptance.

Reviewer #2: Dear Author.

This study aimed to compare linear and non-linear models for fitting the torque-velocity relationship in postmenopausal women and assess how data processing affects the results. 16 physically active postmenopausal women participated in the experiments. The obtained results are well presented and discussed, and therefore I accept the paper.

Reviewer #3: This study presents a valuable investigation into the torque-velocity (TV) relationship in postmenopausal women, offering significant insights into the influence of data processing methods and regression model selection on reported outcomes. While the study designs, protocols, statistical analysis, and corresponding findings are robust, I believe some minor revisions would further enhance its clarity and overall impact.

1. Experimental setup illustrations.

The detailed descriptions of the experimental setup are appreciated. However, incorporating visual aids, such as diagrams or photographs illustrating the participant's positioning and dynamometer attachement for both knee extensor and elbow flexor assesments, would greatly facilitate reader comprehension.

2. Rationale for Polonomial model (PM) selection.

The conclusion to recommend the PM for fitting TV data is well-supported by its superior goodness of fit and the issues identified with other models. To further address this, it would be veneficial to expand on the technical or physiological reasons why the PM is particularly well-suited to reflect the muscle's TV characteristics in this population, beyond just its model performance. A brief discussion of how PM effectively captures the nonlinear aspects of muscle force-velocity relationships, perhaps referencing relevant biomechanical principles or prior research that explains PM's physiological appropriateness, would add greater depth to this key conclusion.

3. The significance of T0 and V0 parameter.

The analysis and comparison of T0 and V0 are central to the study's findings regarding model selection and data processing. While the manuscript defines these parameters according to each model, their fundamental physiological and practical significance within muscle meachnics and why their compraison is crucial could be more explicitly articulated.

In summary, this is a well-conducted study, and the findings are important. I believe that by explicitly discussing the scientific rationale behind the findings, the manuscript could more effectively convey its insights and provide greater value to the readership. I apologize if my understanding was incomplete in certain areas, and I would appreciate any clarifications you could provide to address these points.

6. PLOS authors have the option to publish the peer review history of their article (what does this mean? ). If published, this will include your full peer review and any attached files.

**Do you want your identity to be public for this peer review?** For information about this choice, including consent withdrawal, please see our Privacy Policy .

Reviewer #1: No

Reviewer #2: No

Reviewer #3: No

---

## [Author Response · Author response to Decision Letter 1]

10 Jun 2025

JOURNAL REQUIREMENTS

RESPONSE 1: We have ensured that the manuscript fully complies with PLOS ONE's style requirements

“This study is part of the project PID2021-124277OB-I00 funded by MCIN/AEI/ 10.13039/501100011033 and “ERDF/EU. M.R.A. and J.R.V. received financial support from the Spanish Ministry of Universities through the Grants for the Requalification of the Spanish University System under the Postdoctoral Margarita Salas Programme – Universidade da Coruña (RSUC.UDC.MS09 and RSUC.UDC.MS10, respectively). MRA also acknowledges the financial support received from the Xunta de Galicia (Consellería de Cultura, Educación, Formación Profesional y Universidades) through the Xunta de Galicia Postdoctoral Fellowships (ED481B ‐2024 ‐077). I.N. is supported by a predoctoral grant from Spanish Ministry of Science, Innovation and Universities (FPU23/03727). “

RESPONSE 2: As requested, we have included the following statement at the end of the financial disclosure in the cover letter: "The funders had no role in study design, data collection and analysis, decision to publish, or preparation of the manuscript.".

“This study is part of the project PID2021-124277OB-I00 funded by MCIN/AEI/ 10.13039/501100011033 and “ERDF/EU. M.R.A. and J.R.V. received financial support from the Spanish Ministry of Universities through the Grants for the Requalification of the Spanish University System under the Postdoctoral Margarita Salas Programme – Universidade da Coruña (RSUC.UDC.MS09 and RSUC.UDC.MS10, respectively). MRA also acknowledges the financial support received from the Xunta de Galicia (Consellería de Cultura, Educación, Formación Profesional y Universidades) through the Xunta de Galicia Postdoctoral Fellowships (ED481B ‐2024 ‐077). I.N. is supported by a predoctoral grant from Spanish Ministry of Science, Innovation and Universities (FPU23/03727). Finally, we sincerely thank all the participants in this study for their effort and commitment to the study. We truly appreciate the time, patience, and enthusiasm they have shown in being part of the CARE project.”

“This study is part of the project PID2021-124277OB-I00 funded by MCIN/AEI/ 10.13039/501100011033 and “ERDF/EU. M.R.A. and J.R.V. received financial support from the Spanish Ministry of Universities through the Grants for the Requalification of the Spanish University System under the Postdoctoral Margarita Salas Programme – Universidade da Coruña (RSUC.UDC.MS09 and RSUC.UDC.MS10, respectively). MRA also acknowledges the financial support received from the Xunta de Galicia (Consellería de Cultura, Educación, Formación Profesional y Universidades) through the Xunta de Galicia Postdoctoral Fellowships (ED481B ‐2024 ‐077). I.N. is supported by a predoctoral grant from Spanish Ministry of Science, Innovation and Universities (FPU23/03727).”

RESPONSE 3: As suggested, we have removed the funding statement from the Acknowledgments section.

4. In the online submission form, you indicated that “The data are available on request from the author”

All PLOS journals now require all data underlying the findings described in their manuscript to be freely available to other researchers, either 1. In a public repository, 2. Within the manuscript itself, or 3. Uploaded as supplementary information. This policy applies to all data except where public deposition would breach compliance with the protocol approved by your research ethics board. If your data cannot be made publicly available for ethical or legal reasons (e.g., public availability would compromise patient privacy), please explain your reasons on resubmission and your exemption request will be escalated for approval.

RESPONSE 4: All the data supporting our study’s findings have now been published in a public repository (Zenodo; https://doi.org/10.5281/zenodo.15593957). We have uploaded two datasets in .sav format containing information related to elbow flexor and knee extension exercises. Additionally, we have included these data as supplementary information in .xls format attached to the manuscript. In the online submission form, we have added the following statement: “All the data supporting our study’s findings have been published within the University of A Coruña’s community in Zenodo and are available at the following link: https://doi.org/10.5281/zenodo.15593957. Furthermore, the corresponding .xls files have been included as supplementary information.”

5. When completing the data availability statement of the submission form, you indicated that you will make your data available on acceptance. We strongly recommend all authors decide on a data sharing plan before acceptance, as the process can be lengthy and hold up publication timelines. Please note that, though access restrictions are acceptable now, your entire data will need to be made freely accessible if your manuscript is accepted for publication. This policy applies to all data excepMIt where public deposition would breach compliance with the protocol approved by your research ethics board. If you are unable to adhere to our open data policy, please kindly revise your statement to explain your reasoning and we will seek the editor's input on an exemption. Please be assured that, once you have provided your new statement, the assessment of your exemption will not hold up the peer review process.

RESPONSE 5: Please see our response to Comment 4.

RESPONSE 6: We have updated the reference list to include new articles, and it has been reviewed to ensure compliance with the journal’s formatting requirements

Reviewers' comments:

Reviewer #1: In the article "Exploring the Torque- Velocity Relationship in Postmenopausal Women: Analyzing the Influence of Data Processing" authors discussed the linear and non-linear models for fitting the torque-velocity relationship in postmenopausal women. The article is well supported with faithful and adequate results. Hence, recommended for acceptance.

RESPONSE TO REVIEWER 1: We sincerely appreciate your kind words and valuable feedback. It is truly encouraging to receive such a positive evaluation, and we are pleased that our work has been so well received

Reviewer #2: Dear Author.

This study aimed to compare linear and non-linear models for fitting the torque-velocity relationship in postmenopausal women and assess how data processing affects the results. 16 physically active postmenopausal women participated in the experiments. The obtained results are well presented and discussed, and therefore I accept the paper.

RESPONSE TO REVIEWER 2: Thank you very much for your kind words and thoughtful evaluation. We sincerely appreciate your positive feedback and are grateful for the time and effort you dedicated to reviewing our manuscript

Reviewer #3: This study presents a valuable investigation into the torque-velocity (TV) relationship in postmenopausal women, offering significant insights into the influence of data processing methods and regression model selection on reported outcomes. While the study designs, protocols, statistical analysis, and corresponding findings are robust, I believe some minor revisions would further enhance its clarity and overall impact.

RESPONSE TO REVIEWER 3: Thank you very much for your valuable suggestions and constructive feedback. Your thoughtful comments have helped us refine and improve the manuscript, and we sincerely appreciate your expertise and perspective. Once again, thank you for your support in helping us strengthen our study.

Please find below our responses to each of your specific comments.

1. Experimental setup illustrations. The detailed descriptions of the experimental setup are appreciated. However, incorporating visual aids, such as diagrams or photographs illustrating the participant's positioning and dynamometer attachement for both knee extensor and elbow flexor assesments, would greatly facilitate reader comprehension.

RESPONSE TO REVIEWER 3, COMMENT 1: Thank you for this helpful suggestion. We have added Figure 1, which includes four photographs illustrating the execution of each exercise: elbow flexion and knee extension. The images show both the initial and final positions for each movement, thereby enhancing the clarity of the experimental setup.

2. Rationale for Polonomial model (PM) selection.The conclusion to recommend the PM for fitting TV data is well-supported by its superior goodness of fit and the issues identified with other models. To further address this, it would be veneficial to expand on the technical or physiological reasons why the PM is particularly well-suited to reflect the muscle's TV characteristics in this population, beyond just its model performance. A brief discussion of how PM effectively captures the nonlinear aspects of muscle force-velocity relationships, perhaps referencing relevant biomechanical principles or prior research that explains PM's physiological appropriateness, would add greater depth to this key conclusion.

RESPONSE TO REVIEWER 3, COMMENT 2: This is an excellent point—thank you for the suggestion. We recommend the polynomial model not only due to its superior goodness of fit but also because it provides physiologically coherent parameters. Moreover, it is a simple regression model that is easy to implement for both evaluation and analysis (T = a·V² + b·V + c). The coefficient a represents the second derivative of the function and reflects the curve’s concavity, enabling further exploration. This curvature has been proposed as an indicator of muscle fiber composition, with previous studies reporting that individuals with a higher proportion of fast-twitch fibers tend to exhibit reduced curvature (Tihanyi et al., 1982). In addition, curvature has been suggested as a potential indicator of fatigue, based on changes in force-generating capacity and contraction velocity. Specifically, reductions in both force and maximal shortening velocity lead to decreased power output, resulting in increased curvature in the force-velocity relationship (Jones et al., 2006).

We have included these considerations to reinforce the rationale for selecting this model in the revised version of the manuscript:

Lines 393-398

“Based on our data and the literature, we recommend using the polynomial model due to its strong fit (R² = 0.953–0.999; MSE = 4.02 ± 2.15 N²·m²) and physiologically coherent parameters. Furthermore, the PM is considered a simple regression model that enables efficient parameter extrapolation. Importantly, it allows researchers to analyze the concavity of the torque–velocity curve, which has been proposed as a marker of muscle fiber composition and acute fatigue [28,29].”

Lines 408-410

“Considering the PM’s significantly superior fit (R² = 0.912–0.999; MSE = 1.31 ± 0.93 N²·m²) relative to other models, along with its ability to yield meaningful insights into the curvature of the torque–velocity relationship, we recommend its application in postmenopausal women.”

REFERENCES.

Tihanyi J, Apor P, Fekete G. Force-velocity-power characteristics and fiber composition in human knee extensor muscles. Eur J Appl Physiol Occup Physiol. 1982;48 (3): 331–343. Doi: 10.1007/BF00430223

Jones DA, De Ruiter CJ, de Haan A. Change in contractile properties of human muscle in relationship to the loss of power and slowing of relaxation seen with fatigue. Journal of Physiology. 2006;576 (Pt3): 913–922. doi:10.1113/jphysiol.2006.116343

3. The significance of T0 and V0 parameter.

The analysis and comparison of T0 and V0 are central to the study's findings regarding model selection and data processing. While the manuscript defines these parameters according to each model, their fundamental physiological and practical significance within muscle meachnics and why their compraison is crucial could be more explicitly articulated.

RESPONSE TO REVIEWER 3, COMMENT 3: We thank the reviewer for this insightful comment. T₀ represents the theoretical maximum torque for each exercise, corresponding to the estimated torque produced at zero velocity. In the literature, this parameter is commonly associated with maximum isometric torque (Grbić et al., 2017; Sašek et al., 2022). In our study, we also analyzed the relationship between the directly measured maximum isometric torque (at 90° of elbow flexion and 60° of knee extension) and the theoretical T₀ value derived from each regression model (LM, PM, HM). Thus, T₀ serves as a valuable indicator of maximal isometric force, offering a non-invasive alternative to direct measurement.

V₀, on the other hand, denotes the theoretical maximum unloaded shortening velocity. Historically, this parameter was examined by Hill and Edman to explore its association with sarcomere length and isometric force in vertebrate muscle fibers, although their work focused on single fiber experiments. Today, V₀ is typically estimated rather than directly measured, providing a rapid means of assessing the mechanical characteristics of the muscles involved in a specific exercise. As a theoretical construct, V₀ may help identify the range of velocities potentially achievable (i.e., the angular velocities at which a participant can effectively exert force) within a progressive dynamometer-based evaluation.

These concepts have been incorporated into the Introduction section as follows:

Lines 72-81: “Several mathematical approaches can be applied to torque and velocity data to estimate the maximal capacity of active muscles to produce torque (T0), and power at different angular velocities. T₀ is a useful parameter, commonly associated with maximum isometric torque [11,12], and serves as a reliable indicator of isometric force, offering a non-invasive alternative to direct measurement. However, this association has been scarcely explored in middle-aged women. Furthermore, the theoretical maximum angular velocity (V₀) is an estimated representation of the unloaded shortening velocity of the muscles involved in a specific movement. As a theoretical construct, it may help define the range of velocities potentially achievable (i.e., velocities at which a pa

---

## [Editor Report · Decision Letter 1]

Exploring the Torque- Velocity Relationship in Postmenopausal Women: Analyzing the Influence of Data Processing

PONE-D-25-21537R1

Dear Dr. Iglesias-Soler,

We’re pleased to inform you that your manuscript has been judged scientifically suitable for publication and will be formally accepted for publication once it meets all outstanding technical requirements.

Kind regards,

Hasan Sozen

Academic Editor

PLOS ONE

---

## [Editor Report · Acceptance letter]

PONE-D-25-21537R1

PLOS ONE

Dear Dr. Iglesias-Soler,

I'm pleased to inform you that your manuscript has been deemed suitable for publication in PLOS ONE. Congratulations! Your manuscript is now being handed over to our production team.

Kind regards,

on behalf of

Assoc. Prof. Hasan Sozen

Academic Editor

PLOS ONE